# Is Ferroptosis a Key Component of the Process Leading to Multiorgan Damage in COVID-19?

**DOI:** 10.3390/antiox10111677

**Published:** 2021-10-25

**Authors:** Anna Maria Fratta Pasini, Chiara Stranieri, Domenico Girelli, Fabiana Busti, Luciano Cominacini

**Affiliations:** Department of Medicine, Section of Internal Medicine D, University of Verona, 37134 Verona, Italy; chiara.stranieri@univr.it (C.S.); domenico.girelli@univr.it (D.G.); fabiana.busti@univr.it (F.B.); luciano.cominacini@univr.it (L.C.)

**Keywords:** ferroptosis, COVID-19, multiorgan damage, iron, GSH-GPX4 axis, oxidative stress

## Abstract

Even though COVID-19 is mostly well-known for affecting respiratory pathology, it can also result in several extrapulmonary manifestations, leading to multiorgan damage. A recent reported case of SARS-CoV-2 myocarditis with cardiogenic shock showed a signature of myocardial and kidney ferroptosis, a novel, iron-dependent programmed cell death. The term ferroptosis was coined in the last decade to describe the form of cell death induced by the small molecule erastin. As a specific inducer of ferroptosis, erastin inhibits cystine-glutamate antiporter system Xc-, blocking transportation into the cytoplasm of cystine, a precursor of glutathione (GSH) in exchange with glutamate and the consequent malfunction of GPX4. Ferroptosis is also promoted by intracellular iron overload and by the iron-dependent accumulation of polyunsaturated fatty acids (PUFA)-derived lipid peroxides. Since depletion of GSH, inactivation of GPX4, altered iron metabolism, and upregulation of PUFA peroxidation by reactive oxygen species are peculiar signs of COVID-19, there is the possibility that SARS-CoV-2 may trigger ferroptosis in the cells of multiple organs, thus contributing to multiorgan damage. Here, we review the molecular mechanisms of ferroptosis and its possible relationship with SARS-CoV-2 infection and multiorgan damage. Finally, we analyze the potential interventions that may combat ferroptosis and, therefore, reduce multiorgan damage.

## 1. Introduction

Coronavirus disease 2019 (COVID-19) is a highly contagious infectious disease caused by the novel coronavirus, severe acute respiratory syndrome coronavirus 2 (SARS-CoV-2) [1,2]. The first cases of COVID-19 were documented in Wuhan, Hubei Province of China in December 2019 [3] and since then, the infection spread worldwide. According to the World Health Organization, SARS-CoV-2 infection reached pandemic status on 20 January 2020 [4]. Although many patients are asymptomatic or have mild symptoms such as fever, fatigue and dry cough, a few cases advance to a more severe form of the illness, primarily in older men with comorbidities [5]. Patients with severe infection can suffer from symptoms correlated with lung, heart, kidney, neurological, gastrointestinal and liver injuries as well as from immune and coagulation dysfunction [6]. As angiotensin converting enzyme 2 (ACE2) is widely expressed in these organs [7], SARS-CoV-2 has the potential to intrude on these tissues and damage these organs, leading to multiple organ damage [8].

## 2. SARS-CoV-2 Infection and Multiorgan Damage

Even though respiratory tract infection and the following pulmonary compromise are the main hallmarks of COVID-19, several extrapulmonary manifestations often occur during severe SARS-CoV-2 infection [8]. Evidence indicates that the heart is the second major organ most affected by SARS-CoV-2 entry via the ACE2 receptor, largely present in this organ [7]. SARS-CoV-2 invades myocardial cells probably through pericytes, resulting in increased macrophage infiltration, capillary endothelial cell dysfunction and decreased ACE2 expression [9]. As a matter of fact, an augmented occurrence of cardiovascular (CV) complications has been detected in severe SARS-CoV-2 infection, with a high rate of typical symptoms such as chest pain, shortness of breath, syncope, tachycardia and regional wall motion irregularities or left ventricular malfunction on echocardiography and ST segment or T-wave abnormalities on ECG [3,10,11,12]. The cardiac involvement of COVID-19 is variable and ranges from mildly elevated cardiac biomarkers to acute cardiogenic shock and sudden cardiac death. Thrombi caused by SARS-CoV-2 infection in the coronary area may result in acute coronary syndromes [13]. Systolic right and left ventricular malfunction with or without cardiogenic shock [14] and myopericarditis [15] are also common and may be caused by invasion of the myocardium by SARS-CoV-2 or by an increased inflammatory reaction [10]. In addition, the sustained amplification of inflammatory cytokines, combined with a hypoxic state resulting from pulmonary dysfunction, may cause a fall in coronary blood flow and oxygen supply, which may lead to severe myocardial damages and failure [10]. Several studies have revealed that myocardial injury is common and is significantly associated with fatal outcome of COVID-19 patients, with a significantly higher risk of mortality from the time of symptom onset or hospital admission [10,11,16]. Furthermore, pre-existing CV diseases have also been related to a poor prognosis, since SARS-CoV-2 infection has been shown to worsen cardiac tissue damage and dysfunction [10,17,18,19].

In addition to the respiratory tract and heart injuries, the kidneys are among the most common targets of SARS-CoV-2 [20]. Although the highest levels of SARS-CoV-2 copies per cell are detected in the respiratory tract, data from autoptic series show that SARS-CoV-2 is present in different kidney compartments, mainly in renal parenchyma, glomerular epithelial, endothelial and tubular cells [20]. Clinically, the most frequent kidney damage is acute kidney injury (AKI), but other clinical presentations comprise transient AKI and acute tubular necrosis. The pathogenesis of AKI is considered multifactorial: in fact, besides the direct effects of SARS-CoV-2 via ACE2 widely expressed in different areas of the kidney [21], there are many indirect effects that are associated with secondary kidney damage [22]. For instance, ischemic damage, virus-induced coagulation disorders, or cytokine and complement activation are suggested to play a role in the reduced renal perfusion [22]. Evidence accumulating over time in hospitalized patients with COVID-19 indicates that the incidence of AKI varies greatly [19,23,24,25], reaching even 80% in patients admitted to intensive care unit (ICU). Most of these cases were severe, not transient and nearly 20% required renal replacing therapy (RRT) [23]. In a very large study comprising 5700 hospitalized patients, 20% of the patients developed AKI, of whom only 3.2% necessitate RRT [24]; on the contrary, Cheng et al. [25] reported that only 5% of the hospitalized patients with proteinuria, hematuria and a low glomerular filtration rate thereafter developed AKI. From the first evidence showing that around 50% of the non-survivor patients developed AKI, which was found only in 1% of the survivors [19], several other studies confirmed that SARS-CoV-2-induced AKI is linked with up to five times higher in-hospital mortality than that of patients without AKI [12,17,19,25,26]. During severe SARS-CoV-2 infection, renal dysfunction and AKI often occur in patients with healthy kidneys, but especially in patients with pre-existing chronic kidney disease (CKD) and, in particular, in kidney-transplanted or in RRT patients [22,25,27]. Most importantly, pre-existing CKD has been associated with an increased risk of severe COVID-19 and mortality, particularly in older kidney-transplanted patients [22,25,27]. In addition, a recent study collecting data from more than 17 million patients in the UK and aiming to identify risk factors for COVID-19 mortality indicates that patients with CKD are at higher risk of mortality than those with other known risk factors, including chronic heart and lung disease [27].

Different degrees of liver damage have also been reported in patients with SARS-CoV-2 infection [28,29]; overall, an underlying chronic liver disease was present in 2–11% of hospitalized patients with COVID-19 and 14–53% with COVID-19 developed hepatic dysfunction, particularly those with severe COVID-19 [28,29]. Even though an average expression of ACE2 receptor was found in liver cells [30], it is not clear whether the liver damage is caused by the viral presence, medications given for SARS-CoV-2 infection, or it is multifactorial. Nevertheless, cholangiocytes in bile ducts were reported to express a large number of ACE2 receptors [31], which may favor viral binding and the development of severe post-COVID-19 cholangiopathy with potential for long-term hepatic morbidity [32]. At variance, immune responses to viral infection activated by either immune involvement or collateral injury to cytotoxic T cells and Kupffer cells in the liver may induce liver damage regardless of viral presence in the liver cells [33]. In this context, aspecific inflammatory indicators such as sustained neutrophil count and neutrophil to lymphocyte ratio, coagulation and fibrinolysis pathway activation due to prolonged inflammation, high concentrations of ferritin may indicate the involvement of hepatic cells [34]. However, a phenomenon called ’bystander hepatitis’ that increases aminotransferases and cytokines without damaging liver function may suggest general immune stimulation or inflammation in the course of systemic viral infections [35]. Drug-induced liver damage is a plausible addictive factor to the observed liver blood marker abnormalities after the beginning of therapy [33]. Furthermore, as recently reported, COVID-19 in patients with chronic liver disease (CLD) may precipitate acute-on-chronic liver failure (ACLF) [36]: in-hospital mortality was similar between patients with CLD with or without cirrhosis but was higher in those with cirrhosis who developed ACLF, with a trend for increased mortality by grade of ACLF [36].

In conclusion, observational studies revealed that organ damages are associated with death in the course of SARS-CoV-2 infection [11,19,25,36,37,38]. Overall, the incidence of organ injuries was significantly higher in non-survivor patients than in survivors [19,37,38,39] and resulted to be an independent risk factor of death from COVID-19 [11,25].

## 3. Ferroptosis Signature in SARS-CoV-2 Infection and Molecular Mechanisms of Ferroptosis

As described above, cardiac involvement is a potentially fatal feature in severe SARS-CoV-2 infection [8,9,10,11]. In this context, Jacobs et al. [40] reported on a 48-year-old male patient who died in spite of extracorporeal life support, RRT and maximal pharmacological therapy. Histopathological assessment confirmed the diagnosis of myocarditis with cardiogenic shock. In the lung tissue, diffuse alveolar damage with hyaline formation, marked type 2 pneumocyte hyperplasia with prominent nucleoli and multinucleation as well as interstitial mononuclear inflammatory infiltrates were present [40]. There was also a modest interstitial inflammation in the kidneys and evidence of acute tubular necrosis. Furthermore, liver damage was also detected [40]. The myocardial and renal tissues were also examined for markers of ferroptosis, a novel, iron-dependent nonapoptotic cell death that takes place via disproportionate peroxidation of polyunsaturated fatty acids (PUFAs) in the cell membranes [41]. Very interestingly, immunohistochemical staining of myocardial tissue and proximal tubules of the kidney with monoclonal antibodies binding to oxidized phosphatidylcholine and to 4-hydroxynonenal (HNE), two markers of lipid peroxidation, were positive. These results were the first report of a signature of ferroptosis in COVID-19, which was proposed as a harmful factor in COVID-19 organ damage [40]. Even though, in this patient, it was impossible to stain lung tissue because of multiple lung hemorrhages [40] and further studies are needed to elucidate this specific point, marked production of oxidized phospholipids in the lung tissue had already been shown in the course of SARS-CoV-2 infection [42]. In response to the infection of SARS-CoV-2, iron metabolism dysfunction has been widely documented in a large proportion of COVID-19 patients [43], and this may trigger ferroptosis in the cells of multiple organs.

Since Dixon et al. in 2012 first coined the term of ferroptosis [41] to describe the form of cell death induced by the small molecule erastin, mounting evidence has shown that ferroptosis plays important pathogenetic roles in many CV, liver, lung, kidney, and neurological diseases and has become the focus of investigation to improve their prognosis and treatment [44].

Altered iron metabolism, depletion of glutathione (GSH), inactivation of glutathione peroxidase 4 (GPX4), and upregulation of PUFA peroxidation by reactive oxygen species (ROS) are crucial to the beginning and development of ferroptosis [41,45], which is mechanistically distinct from apoptosis, necroptosis and autophagy [41,45]. In fact, cells undergoing ferroptosis assume a typical rounded form similar to necrotic cells, but there is no cytoplasmic and organelle swelling, or plasma membrane break; characteristically, mitochondria appeared smaller than normal with increased membrane density [41]. Ferroptosis can occur through two major pathways, the extrinsic or transporter-dependent pathway (e.g., decreased cystine or glutamine uptake and increased iron uptake) and the intrinsic or enzyme regulated pathway (e.g., the inhibition of GPX4) [41,45] (Figure 1). Erastin, as a specific inducer of ferroptosis, inhibits cystine uptake in exchange with glutamate by the cystine/glutamate antiporter (system Xc-), thus decreasing cysteine availability, a key precursor of GSH, with consequent malfunction of GPX4 [41] (Figure 1). Iron concentration and distribution in the body is strictly controlled at both the cellular and systemic level [46]. The amount of iron fluxing through the serum iron pool is determined by the rate of iron uptake by cells and the rate of cellular iron release. Normally, as ferrous ion (Fe^2+^) formed by intestinal absorption catalyzes the generation of ROS via the Fenton reaction, it is oxidized by multi-copper iron oxidase hephaestin to ferric ion (Fe^3+^), which binds to transferrin (TF), the major extracellular iron binding protein [47]. Hence, Fe^3+^ bound to TF, which is the principal form of circulating iron, enters cells mainly through membrane TF receptor (TFR) 1 and localizes in endosomes. For the export from the endosomal membrane into the cytosol, Fe^3+^ must be reduced to Fe^2+^ by six-transmembrane epithelial antigen of prostate 3, as the divalent metal-ion transporter (DMT) 1 is a ferrous ion transporter. Iron delivered to the cytoplasm is incorporated into the different iron containing proteins or, if in excess, it is stored in ferritin [47,48]. When needed, Fe^2+^ is exported in the circulation via ferroportin (FPN) [49], which is the only known mammalian protein that exports intracellular iron [50]. Exported Fe^2+^ by cells other than enterocytes is then oxidized to Fe^3+^ by membrane-bound ceruloplasmin, and transported in the circulation predominantly via TF [51]. Several studies show that iron overload due to the imbalance between iron import, storage and export has a key role in influencing the cell susceptibility to ferroptosis [41,45]. In this context, it has been reported that upregulation of the TFR1 promotes iron import, thus triggering ferroptosis [52] and that iron homeostasis is seriously unbalanced in FPN-deficient mice [53]. Since cellular free iron is closely controlled to avoid the generation of ROS, cells have evolved mechanisms whereby iron can be sequestered and released from protein complexes in response to changing iron levels [47]. Excess iron stored in ferritin forms redox-inactive ferritin heteropolymers to preserve tissues and cells against ferroptosis-mediated injury [54]. The nuclear receptor coactivator 4 (NCOA4) has a key role in maintaining intracellular iron homeostasis by facilitating ferritin iron storage or release according to demand. In fact, when cellular iron levels are low, NCOA4-mediated ferritinophagy is induced, thus increasing intracellular iron [55]. Although ferritinophagy is crucial in many iron-dependent physiological processes such as erythropoiesis, recent studies indicate that ferritinophagy may also participate in a ferroptosis mechanism [55] (Figure 1).

The core part of the enzyme-regulated mechanism of ferroptosis is the GSH-GPX4 axis [56] (Figure 1). In fact, GPX4 and its cofactor GSH operate in the protection against ferroptosis by working as reducing agents required for clearance of hydroperoxides (LOOH) generated by ROS [45,56]. As mentioned above, GSH necessitates cysteine for its synthesis from glutamine [57]; cellular import of cystine is coupled to the export of glutamate via system Xc- [57]. Upon hindering this system by the synthetic small molecule erastin or other molecules, the import of cystine is inhibited [41,45]. The resulting cysteine lack impedes synthesis of GSH, and thereby triggers increased glutaminolysis [45]. Disproportionate glutaminolysis stimulates mitochondrial tricarboxylic acid cycle activity and strongly intensifies mitochondrial respiration, leading to hyperpolarization and augmented production of ROS, which ultimately promotes lipid peroxidation [45,56]. Inactivation of GPX4, through depletion of GSH or by using GPX4 inhibitors, ultimately results in uncontrollable lipid peroxidation and cell death through ferroptosis [45,56]. GSH, in fact, is crucial for the GPX4-catalyzed activity as it functions as an electron donor for reducing toxic phospholipid LOOH to nontoxic phospholipid alcohols, and the oxidized GSH (GSSG) is generated as a by-product [58]. GSH can be restored by reducing GSSG using glutathione reductase and nicotinamide adenine dinucleotide phosphate (NADPH) as the electron donor [58,59].

As also shown in Figure 1, PUFAs are a double-edged sword and their peroxidation may damage cells and lead to ferroptosis [60]. PUFAs, as constituents of the cell membranes, control several biological activities, such as inflammation, immunity, synaptic plasticity and cellular growth [61]. The structure of PUFAs is prone to oxidation because the weak C-H bond at the bis-allylic positions is the primary target of ROS attack [62,63]. So far, the machinery starting lipid peroxidation is not entirely established, but can potentially take place through non-enzymatic and/or enzymatic processes [61]. It is likely that non-enzymatic lipid peroxidation is governed by the Fenton reaction, which, through iron and oxygen, triggers the propagation of phospholipid peroxidation and production of phospholipid LOOH [61]. As for the enzymatic mechanism, lipoxygenases and/or cytochrome P450 oxidoreductase have been involved in the process of lipid peroxidation, but a definite connection between these enzymes and the ferroptotic process is still missing [61]. When the initial phospholipid LOOH is generated, it can react with the intracellular labile iron pool to generate alkoxyl and peroxyl radicals, which favor LOOH production and propagation [64]. As a substrate of lipid signaling mediators, free PUFAs should be esterified into membrane phospholipids before acting as ferroptosis inducers [64]. In this context, only a particular class of phospholipids, phosphatidylethanolamines (PEs) which harbor two acyls arachidonoyl (AA) and adrenoyl moieties (AdA) are the preferential substrates of oxidation in ferroptosis [63]. The long-chain AA/AdA can be tied up with coenzyme A by Acyl-CoA synthetase long-chain family member 4 (ACSL4), esterified into PEs by lysophosphatidylcholine acyltransferase-3 (LPCAT3) and, finally, oxidized to LOOH through non-enzymatic and or enzymatic processes [61]. If the LOOH cannot be degraded by GPX4 in time, the excessive lipid peroxides will lead to ferroptosis [64].

## 4. Proposed Mechanisms of SARS-CoV-2 Infection and Ferroptosis Induction

### 4.1. Dysregulation of Iron Metabolism in SARS-CoV-2 Infection

Dysregulation of iron metabolism (i.e., increased plasma levels of ferritin and decreased levels of iron and TF saturation) and anemia are frequently described in COVID-19 patients [24,37,65,66,67,68,69,70,71] and are suggested to play a main role in multiorgan failure of severe SARS-CoV-2 infection. In particular, data from observational studies, which have been confirmed in a large meta-analysis, showed that hemoglobin level decreases, particularly in older people, and is negatively associated with comorbidity and severity of the disease [65,66,68]. Additionally, higher levels of ferritin have been constantly reported in COVID-19 and between non-survivors and survivors, suggesting that mortality might be due to virally driven hyperinflammation [67,68]. Interestingly, anemia, hypoferremia and hyperferritinemia, which are constant features of COVID-19 patients, resemble anemia of inflammation [72]. In this context, Mehta et al. [73] have recently proposed that COVID-19 can be part of the broader spectrum of hyperinflammatory syndromes characterized by cytokine release and hyperferritinemia, one of the key features of these syndromes. It is well established that systemic immune activation leads to profound changes in iron trafficking, resulting in cellular iron retention and in reduced dietary iron absorption [46]. In response to microbial or viral molecules, multiple inflammatory cytokines and, in particular, interleukin-6 (IL-6) are released by cells of the immune system and alter systemic iron metabolism [74]. For viral replication, enhanced cellular metabolism and optimal iron levels within host cells are necessary [75]. In these conditions, IL-6 directly upregulates the expression of liver hepcidin, the master regulator of iron homeostasis, which inhibits the export of iron through FPN, causing cellular iron sequestration [74]. IL-6 also stimulates TFR1 with consequent transferrin internalization and iron accumulation [76]. Moreover, IL-6 increases intracellular ferritin, thus contributing to a decreased iron efflux from the cells [67]. Recent evidence indicates that this scenario may also be applicable to severe SARS-CoV-2 infection, which is characterized by an increase in cytokines, the so-called cytokine storm and especially IL-6 [73,77]. In this context, clinical studies have shown that beyond ferritin, circulating hepcidin levels are also increased in COVID-19 patients and are associated with disease severity [78,79,80]. Therefore, the cellular iron trapping in COVID-19 [65,66] may cause the degradation of ferritin through “ferritinophagy” [81,82] and triggers the increase in labile iron pool that converts phospholipid LOOH to hydroxyl radicals via the Fenton reaction and, eventually, promotes ferroptosis [41] (Figure 2).

### 4.2. GSH-GPX4 Axis in SARS-CoV-2 Infection

GSH is a tripeptide (composed of glycine, cysteine, and glutamate) that can bind to free radicals to preserve cells from oxidative injury [45,56]. GPX4 belongs to a selenoprotein family, whose active sites consist of the amino acid selenocysteine. GPX4 is an important antioxidant enzyme that turns GSH into GSSG and converts the cytotoxic LOOH to the corresponding alcohols. As described above, during ferroptosis, accumulation of redox-active iron depletes GSH pool, which then suppresses the activity of GPX4 and leads to a failing antioxidant response [45,56]. GSH deficiency has already been reported to play a key role in several viral infections, particularly in HIV/AIDS [83], and most recently in COVID-19 [84,85]. In particular, the results of a small study show that decreased GSH was associated with increased ROS and more severe symptoms, whereas in patients with higher levels of GSH, there was a decrease in ROS and a faster recovery from COVID-19 [85]. These results indicate that GSH supplementation may be a useful adjunctive therapy in COVID-19 [86]. Furthermore, in a well-designed study, Wang et al. [87] evaluated the effect of SARS-CoV-2 on the expression of a specific set of selenoprotein in African Green Monkey kidney cells, which express the same 25 human selenoproteins. In particular, they found that SARS-CoV-2 significantly suppressed mRNA expression of ferroptosis-associated GPX4, DNA synthesis-related thioredoxin reductase and endoplasmic reticulum-resident selenoproteins [87]. In summary, the available data so far suggest that SARS-CoV-2 infection is characterized by low GSH pool and downregulation of GPX4 gene expression, both of which facilitate ferroptosis [41].

### 4.3. ROS Generation during SARS-CoV-2 Infection

ROS are oxidants originated by redox reactions, which are considered to be key signals in ferroptosis [88]. As a byproduct of aerobic metabolism, they include superoxide anion, hydroxyl radicals, hydrogen peroxide and singlet oxygen. The main cellular sources of ROS are mitochondrial metabolism [89,90] and nicotinamide adenine dinucleotide phosphate oxidases (NOX) at the cell membrane [91]. Superoxide anion is produced by the electron transport chain on the internal membrane of the mitochondria, and its generation proportion relies on the mitochondrial internal transmembrane potential. Subsequently, superoxide anion and hydrogen peroxide produced by mitochondrial superoxide dismutase (SOD) can propagate from mitochondria to the cytosol [89,90]. The migrated superoxide anion is converted to further hydrogen peroxide by SOD and, eventually, by the Fenton reaction, transformed into the hydroxyl radical, triggering LOOH generation from PUFAs [89,90,92]. In general, viruses can alter mitochondrial dynamics in a highly definite way so that they can successfully replicate [93]. Different mechanisms such as mitochondrial DNA injury, modification of mitochondrial inner membrane potential, alterations in number and distribution of mitochondria, impair antioxidant defense and increased oxidative stress may be implicated [93,94]. While the mechanisms that mitochondria use to defend host cells from SARS-CoV-2 virus have been recently reported [95], current knowledge on how the virus affects mitochondria ROS generation is limited. A previous study on SARS-CoV-2 [96] revealed that open reading frame-9b (Orf9b), one of the accessory proteins of the virus [97], modifies cell mitochondria morphology, interferes with the mitochondrial antiviral signaling system, suppresses interferon (IFN) production and raises autophagy, a cellular machinery activated by ROS. Using affinity purification mass spectrometry, Gordon et al. [98] recently ascertained the presence of a high confidence protein-protein interaction between SARS-CoV-2 Orf9b and mitochondrial translocase of outer membrane (TOM)70. Later on, Jiang et al. [99] showed that SARS-CoV-2 Orf9b binds to TOM70 at the surface of mitochondria membrane and inhibits type I IFN response. In addition, the modified activity of TOM70, by lessening calcium transfer to the mitochondria, diminishes mitochondrial respiration, influences cell bioenergetics and induces the generation of ROS [100] (Figure 3).

As for NOX, many studies have shown that ROS overproduction caused by respiratory viruses is partially mediated by the activity of NOX [101]. In this context, a recent study showed that NOX2 is activated in COVID-19 and is linked with severe clinical outcomes and thrombotic events [102]. The available body of evidence suggests that angiotensin II (Ang II) binding to angiotensin 1 receptor (AT1R) controls the activation of NOX [103,104]. Hence, the SARS-CoV-2-induced ACE2 downregulation increases circulating Ang II and its binding to AT1R, which, by triggering NOX, causes oxidative stress and inflammation in accordance with COVID-19 severity [105] (Figure 3).

SARS-CoV-2 infection is characterized by intracellular iron trapping, inadequate GSH-GPX4 axis and overproduction of ROS, all conditions which may promote LOOH and hydroxyl radical generation via the Fenton reaction and, eventually, promote ferroptosis [41]. So, on the basis of this evidence and of the autoptic evidence in severe SARS-CoV-2 infection [40] and in clinical multiple system diseases [43], we are tempted to speculate that ferroptosis may be an important cause of organ damage in COVID-19 and it might serve as a new adjunctive treatment target.

## 5. Potential Therapeutic Interventions

Since ferroptosis is a regulated cell death controlled by specific intrinsic cellular mechanisms that can be modulated pharmacologically [106], several studies have highlighted ferroptosis as a novel target in many different diseases [44,45,107,108] and, recently, in COVID-19 [40] to prevent organ damage. So far, however, it is not yet established whether ferroptosis inhibition may be helpful in preventing organ damage caused by SARS-CoV-2. Nevertheless, given the increasing recognition of the key role of lipid peroxidation both in SARS-CoV-2 infection and ferroptosis [41,45], there is a strong potential rationale for approaches directed at prevention of lipid peroxidation in clinical setting [107,109]. Potential candidates may be compounds that: strengthen GPX4-GSH axis; induce radical trapping antioxidant (RTA) activity and/or inhibit lipoxygenases (LOXs) and ACSL4 activity; cause cellular labile iron pool depletion; operate through other potential targets.

### 5.1. Compounds That Strengthen the GPX4-GSH-Cysteine Axis

As for GPX4-GSH-cysteine axis, it has been shown that selenium (Se) preserves GPX4 from irreversible inactivation due of its role in selenocysteine synthesis [110]. In addition, it has been recently shown that Se enhances protective transcriptional responses to upregulate GPX4 expression and inhibit ferroptosis in stroke mice model [111]. Intriguingly, in vitro, animal and human studies indicate that Se deficiency is an established risk factor for viral infections and for determining host response: in fact, Se supplementation has been shown to determine important clinical benefits in several viral infections, including HIV-1 [112]. Interestingly, a link between Se status and the outcome of COVID-19 patients has also been identified [113,114,115,116,117]. Overall, the available data so far strongly suggest that Se is essential for prevention of SARS-CoV-2 infection and may negatively impact on COVID-19 outcome particularly in populations where Se intake is low [113].

Relevant to these observations, Ebselen, a synthetic Se compound acting mainly as a peroxiredoxin mimic and as a GPX mimic, was found to have the strongest inhibitory activity against the SARS-CoV-2 main protease required for virus replication [118]. Ebselen, therefore, through its antioxidant activity mediated by restoration of some selenoproteins both reduces viral replication in cell-based assays [118] and inhibit ferroptosis by acting as a GPX mimic [119]. Future research should investigate if this approach is beneficial also in humans.

As cysteine is the rate-limiting precursor for GSH synthesis [45,55], adequate cellular cysteine levels can impede the depletion of GSH, allowing GPX4 to continuously remove lipid LOOH [45,55]. N-acetylcysteine (NAC) is a cysteine prodrug that is utilized in the treatment of liver failure caused by acetaminophen, in critically ill septic patients and to lose thick mucus in chronic obstructive pulmonary disease; more recently, NAC has been used for prevention and as adjuvant therapy in COVID-19 patients [120]. Interestingly, several recent in vitro studies have shown that NAC possesses anti-ferroptotic activity [121,122,123]: in particular, NAC has been demonstrated to hamper cell death in response to erastin-induced ferroptosis, whereas NAC did not exhibit protective effects on cells treated with RAS-selective lethal 3 (RSL3), a compound that can cause ferroptosis through direct binding to GPX4 [121]. Moreover, NAC partially rescued the effect of IL-6 on ROS and ferroptosis in bronchial epithelial cells [122] and the nanoparticle-induced ferroptosis in neuronal cells [123]. NAC, therefore, is likely to work as a precursor of GSH which has anti-ferroptotic roles by enhancing the GPX4-GSH-cysteine axis, than to directly blocking the radical propagation phase as a radical scavenger. Since the results of experimental and clinical studies available so far indicate that NAC acts in a variety of potential therapeutic target pathways involved in the pathophysiology of SARS-CoV-2 infection [124], the fact that NAC inhibits ferroptosis may further explain its adjuvant effect in the treatment of COVID-19 complications.

### 5.2. Compounds That Induce RTA Activity and Inhibit LOXs and ACSL4 Activity

With the accumulating knowledge of the crucial role of lipid peroxidation in ferroptosis and the likely contribution of ferroptosis to degenerative diseases [44] and COVID-19 [40], strategies targeting the suppression of lipid peroxidation have come out as an appealing cytoprotective approach. Since lipid peroxidation can take place via enzymatic and non-enzymatic pathways [61], compounds that can inhibit this process can be divided into two major groups: lipid autoxidation inhibitors or RTAs and LOX inhibitors. The mechanism for ACSL4 inhibitors is basically different from RTAs and LOX inhibitors. As described above, ferroptosis takes places with preferential oxidation of PEs containing AA and AdA [61]; intriguingly, ACSL4 has been recognized as the crucial enzyme for driving AA and AdA to this PE oxidizable pool [125]. In addition, ACSL4-deficient cells are actually resistant to ferroptosis caused by genetic or pharmacological inactivation of GPX4 [125,126]. RTAs, also referred to as chain-breaking antioxidants, are molecules which can quench lipid radicals in nonradical products to block the propagation step [127].

In recent years, ferrostatin-1 (Fer-1) and liproxstatin-1 (Lip-1) have been identified as potent inhibitors of ferroptosis through their strong radical trapping activity able to slow the accumulation of LOOH in PUFAs [128,129]. In comparison with α-tocopherol (α-TOH), evidence indicates that Fer-1 and Lip-1 react significantly more slowly with peroxyl radicals than α-TOH, whereas they are significantly more reactive in phosphatidylcholine lipid bilayers, suggesting the greater potency of Fer-1 and Lip-1 than α-TOH as ferroptosis inhibitors [128]. Lip-1, the first recognized compound of the liproxstatin class, inhibits ferroptosis in the low nanomolar range and shows satisfactory pharmacological characteristics including a short plasma half-life [130]. Significantly, this compound improves acute renal failure in a genetic model of GPX4 deficiency, indicating a strong in vivo anti-ferroptotic activity. Comparably, changes in the native Fer-1 scaffold resulted in second-generation molecules, which grant adequate in vivo stability [131]. Needless to say, Fer-1 and Lip-1 are in their first phase of research in humans and further studies are needed to support the hypothesis they can inhibit ferroptosis in different diseases [44] including COVID-19.

Some years ago, it was hypothesized that LOOH could be produced by PUFAs also through some dioxygenase homologs, namely LOXs [132]. In addition, it was later shown that some of the compounds that inhibit LOXs also possessed RTA properties [133]. However, LOX inhibitors with RTA properties, such as NDGA (pan-LOX inhibitor), zileuton (5-LOX inhibitor) and PD146176 (15-LOX-1 inhibitor), faintly hampered ferroptosis in cell lines overexpressing LOXs compared with Lip-1 and Fer-1 [133]. Among the LOX inhibitors lacking RTA properties, 15-LOX-1 inhibitor ML351, but not 5-LOX inhibitors CAY10649 and CJ-13610 [133,134], significantly inhibit cell death in the RSL3 model [135], indicating a particular role for 15-LOX-1 inhibitors in counteracting ferroptosis. In this context, it was reported that membrane PUFA-containing PEs are the better substrates of 15-LOX-1 to generate harmful LOOH and PE-binding protein 1 produces complexes with 15-LOX-1 to boost the oxidation of AA-PEs [136]. Overall, the hypothesis that LOXs participate in the execution of ferroptosis is still controversial and, therefore, to date, LOX inhibition does not represent a concrete target to suppress ferroptosis [133,134].

As for ACSL4 inhibitors, it was previously shown that peroxisome proliferator-activated receptor γ (PPAR γ) agonists, thiazolidinediones (TZDs), can selectively inhibit ACSL4 over other ACSL isoforms [137]. TZDs including rosiglitazone (ROSI), pioglitazone, and troglitazone demonstrated significant suppressive effects in ferroptotic cell models [134]. Although TZDs have been promoted as insulin sensitizers acting as PPAR γ agonists, their effect on ferroptosis was attributed only to their inhibition of ACSL4 [107]. A reduction in AA- and AdA-containing PEs was noticed in both ACSL4-KO cells and ROSI-treated wild type cells [134]; in addition, ROSI remarkably extended the survival of GPX4-KO mice compared with controls [134], indicating that TZDs lessened ferroptotic damage by reducing lipid peroxidation and increasing GSH and GPX4 levels. ROSI was also shown to inhibit ferroptosis following ischemia reperfusion [138,139]. Taken together, this indicates that ACSL4 may be a promising target for different ferroptosis-relevant pathological process.

### 5.3. Compounds That Induce Cell Iron Depletion

It is known that SARS-CoV-2 requires iron for its replication and functions [140]. While the increase in ferritin production on the one hand allows adequate storage of iron and reduces iron damage [141], on the other hand, it can be associated with increased ferritinophagy [81,82]. This implies a rise in labile pool iron [81,82], which converts LOOH to hydroxyl radicals via the Fenton reaction and, eventually, promotes ferroptosis [41]. Thus, since there are many iron chelating agents currently used in clinical practice, iron chelation therapy may represent a useful therapeutic approach to COVID-19. Deferoxamine (DFO), an injective iron chelator, is a polar compound with poor membrane permeability, which can enter the cells through endocytosis [142]. It reacts with Fe^3+^ and produces a fixed octahedral coordination product, feroxamine, which can be removed by the kidneys [142]. DFO also directly promotes lysosomal autophagy of ferritin [143]. Since DFO was the first molecule shown to suppress ferroptosis induced by erastin and RSL3, it may be considered the drug of choice to prevent lipid peroxidation and ferroptosis under different pathological conditions [41]. Deferasirox is a membrane-permeable oral iron chelator with a tridentate ligand selective for Fe^3+^ to form a stable complex, which is excreted through the kidneys [142]. Deferasirox has been shown to chelate both cytosolic iron and iron, which is extracted from ferritin before ferritin degradation by proteasomes [143]. Deferasirox has been reported to inhibit hemin-induced ferroptotic cell death and ROS generation in human monocytes [144].

Deferiprone is an oral iron chelator that binds Fe^3+^ to form a stable complex, which is then excreted in the urine [142]. Compared to DFO and deferasirox, deferiprone is most useful in the chelation of iron in the heart and similar to DFO in the chelation of liver iron [142]. Interestingly, deferiprone was reported to specifically inhibit ferroptosis induced by erastin and glutamate in human mesencephalic cells [145].

Baicalein is a flavonoid that possesses hydroxyl groups forming complexes with iron in a stoichiometry of 1:1 [142]. Baicalein has been shown to exhibit a marked protection against erastin-induced ferroptotic cell death in human pancreatic cancer cell lines and to reverse erastin-induced intracellular iron accumulation, GSH depletion, and GPX4 degradation [146]. In addition to its iron chelator activity, baicalein also interferes with the activity of several LOXs, which may also contribute to its anti-ferroptotic action [135].

2,2′-bipyridine is another iron chelator whose characteristic is to specifically sequester iron from labile pool iron in cells [142]. The 2,2′-bipyridine may also move into mitochondria and bind mitochondrial iron, therefore reducing the production of ROS [147]. As anticipated, 2,2′-bipyridine inhibits ferroptosis by decreasing iron-dependent lipid peroxidation [41].

### 5.4. Compounds That Operate through other Potential Targets

Recent evidence points to a key role of ferroptosis suppressor protein 1 (FSP1) in inhibiting ferroptosis, by directly reducing the oxidized endogenous molecule coenzyme Q10 (CoQ10) with NADH as a cofactor [148]. Cancer cell lines stably expressing FSP1 were reported to be preserved from ferroptosis, including from that caused by GPX4 inhibitors [148] and reduced CoQ10 was assessed to work as a lipophilic RTA in the plasma membrane [149]. In this context, it was shown that idebenone, a hydrophilic reduced analog of CoQ10, could inhibit FIN56-induced ferroptosis, while the supplementation of CoQ10 was found to be ineffective owing to its high hydrophobicity [150]. Idebenone has already been assessed in several preclinical studies and clinical trials related to mitochondrial and neurological diseases, owing to its antioxidant properties [134], but the results are uncertain and further studies are needed to confirm that idebenone is of use in contrasting organ damages caused by ferroptosis.

GTP cyclohydrolase 1 (GCH1), which catalyzes the rate-limiting step in the biosynthesis of antioxidant tetrahydrobiopterin (BH4), showed selective cytoprotective activity against ferroptosis irrespective of the GPX4-GSH axis [151]. Supplementation with BH4 or BH2 (a partially oxidized derivative of BH4) prevented ferroptosis in a dose-dependent manner in several cell lines [151]. It was recently demonstrated that dihydrofolate reductase (DHFR) is a negative regulator of ferroptosis by regenerating oxidized BH4. Pharmacological suppression or genetic deletion of DHFR cooperated with GPX4 inhibition to induce ferroptosis [152]. All together, these results suggest that targeting the GCH1-BH4-DHFR axis offers a new policy for controlling the cellular sensitivity to ferroptosis.

Finally, Nuclear factor erythroid 2 p45-related factor2 (NRF2) is a well-known transcription factor that plays a key role against oxidative stress [153]. Under basal conditions, Nrf2 binds to Kelch-like enoyl-CoA hydratase-associated protein1 (Keap-1) and continues to be inactivated by ubiquitination and proteasomal degradation [153]. When cells are exposed to oxidative stress, NRF2 is released from the Keap1 and quickly relocated into the nucleus, where it interacts with the antioxidant response element in the promoter region of the target gene and then switches on the transcriptional pathway to counterbalance oxidative stress and preserve cellular redox homeostasis [153]. As there is evidence that NRF2 regulates ferroptosis in various ways, the relation between NRF2 and the pivotal pathways for ferroptosis has, by this time, attracted much attention [154]. In particular, analyzing the networks of ferroptosis-related proteins targeted by NRF2 through the STRING database website and Cytoscape 3.7.1 software analysis [155], it was shown that NRF2 can directly control ferroptosis through GPX4 synthesis-related enzyme (i.e., glucose-6-phosphate dehydrogenase, GSH reductase, GPX4, glutamate-cysteine ligase modifier subunit,12-channel transmembrane protein transporter vector family 7 member 11, glutamate-cysteine ligase catalytic subunit, thioredoxin reductase 1 or through the PPAR γ pathway) [155]. Otherwise, it can indirectly regulate ferroptosis by controlling intracellular iron concentration via NRF2-heme oxygenase 1 (HO-1)–iron regulatory-related protein axis (i.e., biliverdin reductase A/B, ferritin heavy chain 1, transferrin receptor 1, recombinant ferrochelatase and FPN) [154]. In summary, all the numerous proteins involved in the ferroptosis process and that can roughly be classified into four groups, i.e., GPX4 synthesis and function-related, iron metabolism-related, lipid peroxidation-related and transcription factor-related, can be mediated by NRF2 target genes [154,156]. Interestingly, the NRF2 pathway was found to be suppressed in lung biopsies of patients affected by SARS-CoV-2 [157] and it has been suggested that NRF2 activators may be a potential adjuvant therapy against SARS-CoV-2 infection [158]. Since NRF2 controls ferroptosis and ferroptosis may take part in organ damages during COVID-19 infection [40,44], targeting NRF2 with NRF2 activators as already suggested for combating SARS-CoV-2 infection [124], is expected to become a new direction for the prevention of these complications.

Increasing evidence demonstrates that natural compounds, such as saponins, flavonoids and isothiocyanates, can either induce or inhibit ferroptosis [159]. In particular, flavonoids, polyphenols and phenylpropanoids have been shown to regulate Fe^2+^ metabolism, inhibit LOXs and ROS, as well as activate NRF2/GPX4 signaling to antagonize ferroptosis [159]. With the rapid growth of works on ferroptosis, the interaction between ferroptosis and natural products has been gradually established. The results from the studies available so far [159] clearly indicate the need to also test bioactive phytochemicals in randomized clinical trials to combat ferroptosis and, therefore, reduce multiorgan damage.

## 6. Conclusions

In conclusion, multiorgan damage often occurs during exacerbations of SARS-CoV-2 infection [8]. Several studies have highlighted ferroptosis as a novel therapeutic target in different diseases [44,45,107,108] and possibly in COVID-19 [40] to prevent organ damage. So far, it is not yet established if ferroptosis inhibition may be helpful in the treatment of organ damage produced by SARS-CoV-2 infection. However, the fact that intracellular iron trapping [47,65] depletion of GSH, inactivation of GPX4 [80,81,82,83], and upregulation of PUFA peroxidation [40] are constitutive components of both SARS-CoV-2 infection and ferroptosis [41], suggests that highly potent ferroptosis inhibitors may be in first position for clinical validation as adjuvant therapy. Drugs that potentiate the GPX4-GSH axis, induce RTA and ACSL4 activity and, finally, cause labile pool iron depletion are likely candidates in COVID-19 treatment. In addition, the fact that Nrf2 directly or indirectly regulates antioxidant capacity and the HO-1–iron regulatory-related axis [154,155] indicates that NRF2 activators may become a new approach for the treatment of organ damage in COVID-19. More attention and studies are needed in the next future to confirm the role of ferroptosis in the pathogenesis of organ damage in COVID-19 and to look at the correct therapy for its prevention.

## Figures and Tables

**Figure 1 antioxidants-10-01677-f001:**
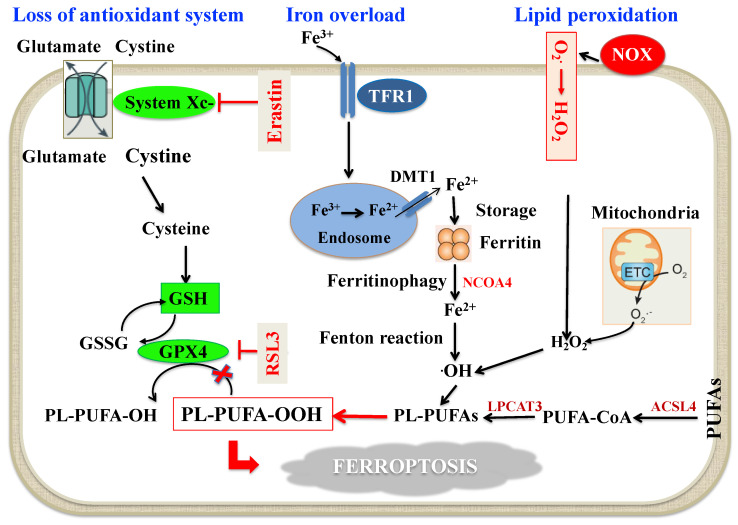
Schematic molecular mechanisms of ferroptosis. *Loss of antioxidant system*: System Xc- facilitates the cellular uptake of cystine, which is reduced by cystine reductase to cysteine, a key precursor in GSH synthesis. GSH serves as an important reducing cofactor for GPX4 (which catalyzes the reduction of lipid peroxides to their corresponding alcohols) while also forming GSSG. Erastin inhibits the cellular uptake of cystine, thus impairing intracellular GSH synthesis. Depletion of GSH leads to the indirect inactivation of GPX4, and the resulting accumulation of lipid peroxides disrupts membrane integrity resulting in ferroptosis. RSL3 acts by directly inactivating GPX4 and does not interfere with the cellular uptake of cystine or intracellular GSH synthesis. *Intracellular iron accumulation*: Circulating iron (Fe^3+^) bound to transferrin enters into cells by TFR1, localizes in endosomes where it is deoxidized to Fe^2+^. Ultimately, Fe^2+^ is released into a labile iron pool in the cytoplasm by DMT1, while excess iron is stored in ferritin. Under certain circumstances, there may be degradation of ferritin through autophagy, which is termed as “ferritinophagy”, a phenomenon that triggers an increase in labile iron pool and ·OH via the Fenton reaction. This process requires H_2_O_2_ production by activation of NOX, or mitochondria ETC pathways. *Lipid peroxidation*: The generation of PL-PUFAs by ACSL4 and LPCAT3 have a main role in promoting lipid peroxidation. At the final step of ferroptosis, lipid peroxidation directly or indirectly induces pore formation of cell membrane, thus triggering cell death. Abbreviations: ACSL4: Acyl-Coenzyme A synthetase long-chain family member 4; DMT1: Divalent metal transporter 1; ETC: Electron transport chain; GPX4: Glutathione peroxidase 4; GSH: Reduced glutathione; GSSG: Oxidized glutathione; H_2_O_2_: Hydrogen peroxide; LPCAT3: Lysophosphatidylcholine acyltransferase 3; NCOA4: Nuclear receptor coactivator 4; NOX: Nicotinamide adenine dinucleotide phosphate oxidase; O2^−^: Superoxide; ·OH: Hydroxyl radical; PL-PUFA-OH: Phospholipid polyunsaturated fatty acid alcohols; PL-PUFA-OOH: Phospholipid polyunsaturated fatty acid peroxides; PUFA-CoA: Polyunsaturated fatty acid coenzyme A; RSL3: RAS-selective lethal 3; TFR1: Transferrin receptor 1.

**Figure 2 antioxidants-10-01677-f002:**
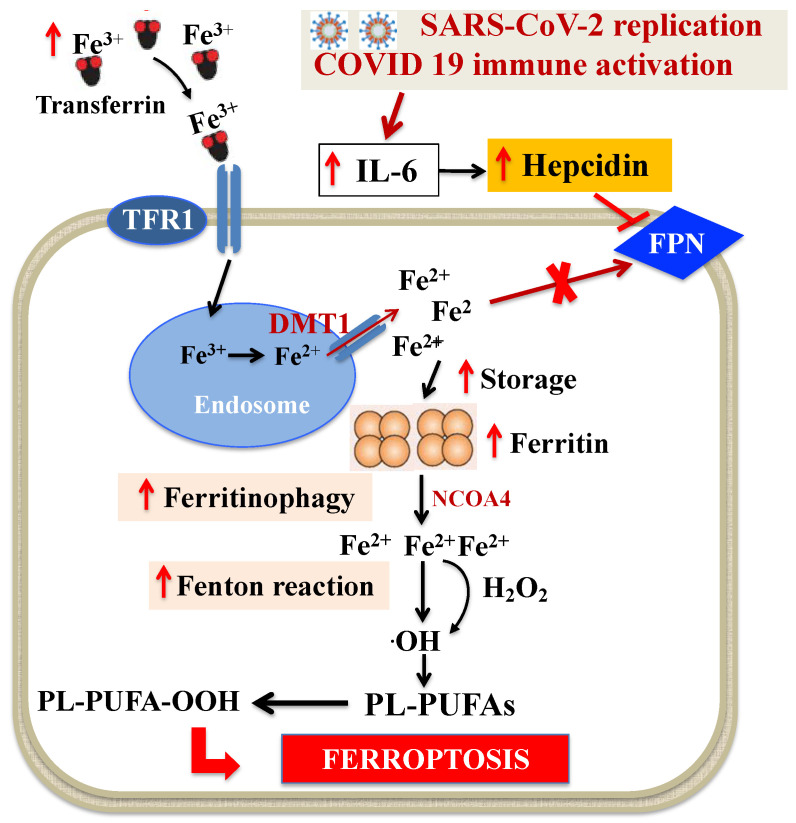
Dysregulation of intracellular iron metabolism in SARS-CoV-2 infection. Systemic immune activation induced by SARS-CoV-2 leads to profound changes in iron trafficking, resulting in cellular iron retention. Fe^3+^ bound to TF is the principal form of circulating iron, which enters cells through membrane TFR1 and localizes in endosomes, where the ferrireductase activity of six transmembrane epithelial antigen of the prostate 3 reduces Fe^3+^ to Fe^2+^. Finally, DMT1 releases Fe^2+^ from endosomes, thereby increasing the concentration of labile iron pool, while excess iron is stored in ferritin. SARS-CoV-2-related huge increase in cytokines, and especially IL-6, stimulates hepcidin which, in turn, blocks FPN, causing the increase in intracellular iron and of ferritin. This iron retention can cause ferritinophagy and triggers the increase in labile iron pool that induces ·OH via the Fenton reaction and, eventually, through PL-PUFA peroxidation promotes ferroptosis. Abbreviations: DMT1: Divalent metal transporter 1; FPN: Ferroportin; H_2_O_2_: Hydrogen peroxide; IL-6: Interleukin-6; NCOA4: Nuclear receptor coactivator 4; ·OH: Hydroxyl radical; PL-PUFAs: Phospholipid polyunsaturated fatty acids; PL-PUFA-OOH: Phospholipid polyunsaturated fatty acid peroxides; TF: Transferrin; TFR1: Transferrin receptor 1.

**Figure 3 antioxidants-10-01677-f003:**
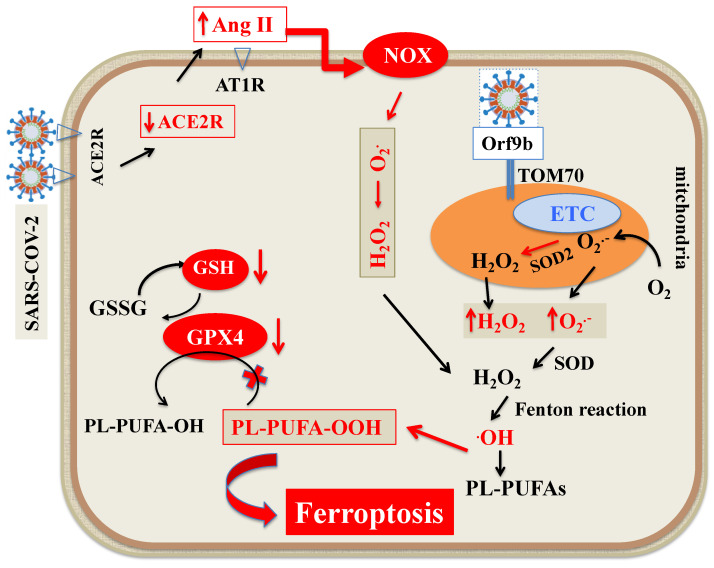
Scheme of ROS generation during SARS-CoV-2 infection. The main cellular sources of ROS are mitochondria and NOX at the cell membrane. O_2_^●−^ is produced by ETC on the internal membrane of the mitochondria. Subsequently, O_2_^●−^ and H_2_O_2_ produced by mitochondrial SOD2 can propagate from mitochondria to the cytosol. The migrated O_2_^●−^ is converted to further H_2_O_2_ by SOD and eventually, by Fenton reaction, transformed into ·OH, triggering LOOH generation from PUFAs. In addition, Orf9b, one of the accessory proteins of SARS-CoV-2 increases ROS generation by binding to TOM70 at the surface of mitochondria membrane. As for NOX, SARS-CoV-2 by downregulating ACE2 receptors increases circulating Ang II and its binding to AT1R, which, by triggering NOX, increases ROS formation. Abbreviations: ACE2R: Angiotensin converting enzyme 2 receptor; Ang II: Angiotensin II; AT1R: Angiotensin 1 receptor; ETC: Electron transport chain; H_2_O_2_: Hydrogen peroxide; LOOH: Peroxides; NOX: Nicotinamide adenine dinucleotide phosphate oxidase; O_2_^●^^−^: Superoxide; ^●^OH: Hydroxyl radical; Orf9B: Open reading frame-9b; PUFAs: Polyunsaturated fatty acids; PL-PUFA: Phospholipid polyunsaturated fatty acids; PUFA-OH: Phospholipid polyunsaturated fatty acid alcohols; PL-PUFA-OOH: Phospholipid polyunsaturated fatty acid peroxides; ROS: Reactive oxygen species; SOD: Superoxide dismutase; TOM70: Translocase of outer membrane 70.

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
