# Peer review of "Is Ferroptosis a Key Component of the Process Leading to Multiorgan Damage in COVID-19?"

_antioxidants, 2021, doi:10.3390/antiox10111677_

Round 1
Reviewer 1 Report
The authors of this manuscript (Manuscript ID: antioxidants-1430035) provide a comprehensive review focused on the molecular mechanisms that trigger ferroptosis and discuss the role of ferroptosis in the pathogenesis of organ damage in COVID-19. The potential therapeutic interventions to combat ferroptosis and reduce multiorgan damage is also analyzed. The topic is interested and covers existing knowledge in literature.
Minor comment:
Line 343-346: The authors explain how hydroxyl radicals are formed in the mitochondria by Fenton reaction. They also state that these hydroxyl radicals propagate from mitochondria to the cytosol triggering LOOH generation from PUFAs.
However, hydroxyl radicals are so reactive that will oxidize molecules at the vicinity of their generation, making unlikely to propagate from mitochondria to the cytosol.
Please rephrase or elaborate on the trigger of LOOH generation from PUFAs.
Author Response
You are right and we thank you very much for rising this important point. Actually the hydroxyl radicals are likely to derive from the hydrogen peroxide and in part also from the anion superoxide generated in mitochondria; these molecules can cross the mitochondrial membrane and migrate into cytoplasm (see new reference 90 that replaces the old reference 90) where eventually, by Fenton reaction, can be transformed into hydroxyl radicals triggering LOOH from PUFAs. Consequently, as you indicated, we rephrased the text from line 343 to line 348 of the revised manuscript and accordingly changed the Figure 3 and its legend.
Reviewer 2 Report
In the present review “Is ferroptosis a key component of the process leading to multiorgan damage in COVID-19?”, Anna Maria Fratta Pasini and colleagues, examined the molecular mechanisms of ferroptosis and its possible relationship with SARS-CoV-2 infection and multiorgan damage. Moreover, the authors analyzed the potential interventions which may combat ferroptosis and therefore reduce multiorgan damage. Overall, I think that the manuscript is well-written (within the scope of this journal), well-structured and the data are of clinical relevance on a current topic of interest. I would like to congratulate the authors on their work. I have a small suggestion/curiosity to improve the quality of review.
In light of the results here obtained, please to discuss on the possible application/role of healthy dietary behaviour and/or nutraceutics, antioxidant, antinflammatory compounds that, acting on molecular signaling pathway directly/indirectly explored in the present paper, could provide a possible strategy to combat ferroptosis and therefore reduce multiorgan damage.
Author Response
Thank you for your suggestion. In order to comply with your request we briefly discussed on the possible utilization of bioactive phytochemicals to combat ferroptosis and therefore reduce multiorgan damage. To support this possibility we added a recent comprehensive review (new reference 160).